# The Effect of Cultivation Passaging on the Relative Telomere Length and Proliferation Capacity of Dental Pulp Stem Cells

**DOI:** 10.3390/biom11030464

**Published:** 2021-03-20

**Authors:** Nela Pilbauerova, Tomas Soukup, Tereza Suchankova Kleplova, Jan Schmidt, Jakub Suchanek

**Affiliations:** 1Department of Dentistry, Charles University, Faculty of Medicine in Hradec Kralove and University Hospital Hradec Kralove, 500 05 Hradec Kralove, Czech Republic; nela.pilbauerova@lfhk.cuni.cz (N.P.); tereza.suchankova@lfhk.cuni.cz (T.S.K.); suchanekj@lfhk.cuni.cz (J.S.); 2Department of Histology and Embryology, Charles University, Faculty of Medicine in Hradec Kralove, 500 03 Hradec Kralove, Czech Republic; soukupto@lfhk.cuni.cz

**Keywords:** telomere, telomerase, dental pulp stem cells, qPCR, relative telomere length measurement

## Abstract

Telomeres are repetitive nucleoprotein DNA sequences that shorten with each cell division. The stem cells activate telomerase to compensate for the telomere loss. This study aimed to evaluate the effect of cultivation passaging on the relative telomere length and proliferation capacity of dental pulp stem cells. We used ten dental pulp stem cell (DPSC) lineages stored for 12 months using uncontrolled-rate freezing to reach the study’s goal. We analyzed their proliferation rate, phenotype using flow cytometry, multipotency, and relative telomere length using a qPCR analysis. We determined the relative telomere length in the added study by performing analysis after one, two, and three weeks of cultivation with no passaging. We documented the telomere attrition with increasing passaging. The shorter the relative telomere length, the lower reached population doublings, and longer population doubling time were observed at the end of the cultivation. We observed the telomere prolongation in DPSCs cultivated for two weeks with no passaging in the added subsequent study. We concluded that excessive proliferation demands on DPSCs during in vitro cultivation result in telomere attrition. We opened the theory that the telomerase might be more efficient during cell cultivation with no passaging. This observation could help in preserving the telomere length during ex vivo DPSC expansion.

## 1. Introduction

Dental pulp stem cells (DPSCs) are mesenchymal adult stem cells. The minimal criteria for mesenchymal stem cells (MSCs) involve [1]:an adhesion to cultivation plastic dish under standard in vitro conditions;an immunophenotype including a cluster of differentiation (CD) markers for MSCs and lacking the expression of CD markers for hematopoietic stem cells;an ability to differentiate into adipogenic, chondrogenic, and osteogenic mature cell populations.

One of the remarkable features of stem cells, including DPSCs, is their self-renewal potential. However, the lifespan of the adult stem cell is limited, and during an in vitro culture, they lose differentiation potential and lower the replicative capacity (replicative senescence) [2]. The replicative senescence was described by L. Hayflick more than 50 years ago [3]. He quantitatively examined the finite lifetime of diploid cells in vitro. Each cell within the population was endowed with the same doubling potential (50 ± 10). Many recent studies have reported that the telomere length shortening is a hallmark of cell senescence, and the maintenance of their length is essential for cell self-renewal ability and differentiation potential [4]. 

Telomeres are nucleoprotein repetitive DNA sequences at the end of a chromosome. They consist of a tandem TTAGGG repeat [5]. They maintain the cell genome’s stability and protect the chromosome’s end from deterioration or fusion with other adjacent chromosomes [6]. During each cell division, the telomeres shorten and, at some point, reach their critical lengths when cell deaths are triggered. A compensatory mechanism counteracts the telomere shortening. The stem cells, cancer cells, and germ cells can activate a ribonucleoprotein enzyme called telomerase to compensate for the telomere loss. The core-complex of telomerase comprises reverse transcriptase hTERT and the RNA component hTR, used as a template to synthesize telomeric DNA [7]. A homeostasis between the telomere length and the telomerase activity has a vital role in embryogenesis [8]. Later, it dictates the proliferative potential of a pool of stem cells and tissue regeneration and reparation in mature organisms [9]. As is described above, each cell division comes at the cost of shortening the telomere length.

Unfortunately, it has been shown that the future potential use of MSCs in regenerative or reparative medicine requires cell amplification in large-scale production via long-term cultivation. Despite the successes that have been achieved so far, several problems in the in vitro cultivation of dental pulp stem cells remain. The most challenging one is maintaining the unchanged proliferation capacity throughout the entire cultivation and keeping an abundant cell source for possible future cell-based therapies. Most clinical protocols recommend 20–100 × 10^6^ hMSCs per treatment; therefore, to obtain such a high number of cells, hMSCs must be expanded in vitro for at least 4–8 weeks before transplantation [10]. Mokry and al. [11] concluded that extensive in vitro proliferation of human DPSCs was associated with telomere attrition with a significant correlation to prolonging DPSC population doubling time (PDT). They determined the telomere length as a biomarker indicating the replicative age of stem cells. If the telomerase had sufficient time, would its compensatory effect have been efficient despite the excessive ex vivo expansion?

The study aimed to evaluate the effect of cultivation passaging on the relative telomere length and proliferation capacity of human dental pulp stem cells (DPSCs). The subsequent pilot study aimed to evaluate the relative telomere length in human DPSCs cultivated in vitro with no passaging. We hypothesized that if the telomerase had enough time to compensate for telomere loss caused by excessive cell amplification in vitro, the telomerase compensatory effect would be more effective.

## 2. Materials and Methods

### 2.1. Human DPSCs

All human DPSC lineages used in this study had been previously isolated and characterized according to their phenotype and differentiation potential. The results had already been published in our previous study [12]. Briefly, all permanent teeth were collected from ten donors at the Dental clinic of the University Hospital Hradec Kralove, Czech Republic. All donors were healthy young individuals aged between 13 and 18 years, with no history of smoking or radiotherapy in the orofacial region. The Ethical Committee in Hradec Kralove, Czech Republic, approved guidelines for this study (ref. no. 201812 S07P). The patients or their legally authorized representatives in the case of underaged donors were briefly informed about the study’s purpose and signed a written informed consent before being included in the study.

We used the ten lineages of DPSCs that were cryopreserved in the 1st passage using an uncontrolled freezing rate and a dimethyl sulfoxide (DMSO, Sigma-Aldrich, Merck KGaA, Germany) in a concentration of 10% as a cryoprotective agent. The median of reached population doublings was 20.0 in the 1st passage. The DPSCs were stored at −80 °C.

At the beginning of this study, we thawed all ten lineages using a 37 °C thermal bath. The thawed DPSCs were expanded in the modified cultivation media (α-MEM Medium, Gibco, Thermo Fisher Scientific, Foster City, CA, USA) for mesenchymal adult progenitor cells, containing 2% fetal bovine serum (FBS, PAA Laboratories, Dartmouth, MA, USA) and supplemented with 10 ng/mL epidermal growth factor (PeproTech, London, UK), 10 ng/mL platelet-derived growth factor (PeproTech), 50 mM dexamethasone (Bieffe Medital, Grossoto, Italy), 0.2 mM l-ascorbic acid (Bieffe Medital), 2% glutamine (Invitrogen, Waltham, MA, USA), 100 U/mL penicillin (Invitrogen), 100 µg/mL streptomycin (Invitrogen), 20 μg/mL gentamicin (Invitrogen), and 2.5 µg/mL amphotericin (Sigma-Aldrich). The medium was also enriched with 10 μL/mL Insulin-Transferrin-Selenium-Sodium supplement (ITS, Bieffe Medital) to increase the nutrient utilization. We incubated the cultivation dishes with adherent stem cells at 37 °C and 5% CO_2_. The cultivation medium was changed every three days. The cells were passaged right before the plateau, when cells were still growing exponentially (70% confluence on average) using 0.05% trypsin-EDTA (Gibco), and then they were reseeded in a concentration of 5000 cells per cm^2^ of an adherent tissue culture dish (TPP, Sigma-Aldrich). The cell confluence was measured using the CKX-CCSW Confluency Checker (Olympus, Tokyo, Japan). We did not perform biological replication of each independent cell lineage. All lineages were terminated in the 8th passage.

### 2.2. DPSC Phenotype Analysis

Since all lineages of DPSCs had already been phenotypically characterized [12], we analyzed just a reduced number of CD markers in the 3rd and 7th passages. We analyzed cryopreserved DPSCs for CD29 (TS2/16, BioLegend, San Diego, CA, USA), CD31 (MBC 78.2, Invitrogen), CD34 (581 (Class III), Invitrogen), CD44 (MEM 85, Invitrogen), CD45 (HI30, Invitrogen), CD73 (AD2, BD Biosciences Pharmingen, Erembodegen, Belgium), CD90 (F15-42-1-5, Beckman Coulter, Miami, FL, USA), CD105 (SN6, Invitrogen) and CD271 (ME20.4, BioLegend) using a flow cytometry analyzer (Cell Lab Quanta, Beckman Coulter). The phenotype analysis was done by first detaching adherent stem cells using the 0.05% trypsin-EDTA solution (Gibco), and then we stained them with primary immunofluorescence antibodies conjugated with phycoerythrin (PE) or fluorescein (FITC). Positive cells were determined as the percentage with a fluorescence intensity greater than 99.5% of the negative isotype immunoglobulin control. The expression of surface markers was classified according to previously established criteria by Suchanek and al. [13]. Classification criteria: <10% no expression, 10–40% low expression, 40–70% moderate expression, and >70% high expression.

### 2.3. DPSC Kinetics

We measured the total cell count in each passage using a Z2-Counter (Beckman Coulter). The proliferation activity was determined as cumulative population doublings (PDs) and population doubling time (PDT). We used the formula PD = log_2_ (N_x_ / N_1_) to calculate the population doublings reached in each passage. N_x_ is the total passage cell count calculated using the Z2-Counter, and N_1_ is the initial cell count seeded into the culture dish (5000 cells / cm^2^). To calculate the population doubling time, we used the formula PDT = *t* / *n*, where t is the number of hours of cultivation per passage, and *n* is the number of PDs in that passage, calculated as described above.

### 2.4. Quantitative Polymerase Chain Reaction (qPCR) Analysis

This study mainly aimed to observe the effect of ex vivo cultivation passaging of the relative telomere length. Therefore, we measured the change of the relative telomere length between the 2nd and 7th passage. We extracted the DNA of isolated stem cells using a DNeasy Tissue Kit (Qiagen, Hilden, Germany) and followed the manufacturer’s instructions. The qPCR assay performed the relative telomere length measurement according to a method described by Cawthon [14] with small modifications. After the DNA isolation, we calculated its concentration in each sample using a spectrophotometer Nanodrop 1000 (Thermo Fisher Scientific, Waltham, MA, USA). The telomere repeat copy number to single-gene copy number (T/S) ratio was determined using the equation: T/S = 2^−ΔCt^ (where ΔCt = C_ttelomere_ − Ct_single-copy gene_). The T/S for each sample was normalized to the T/S value of a reference DNA sample to standardize between different runs. That is, −ΔΔCt was calculated for each sample. This value is proportional to the average telomere length of the evaluated sample. The single-copy gene (housekeeping gene) was a coding acidic ribosomal phosphoprotein 36B4, often used as an internal standard for analysis. We performed the qPCR in 96-well plates, and we analyzed each sample in triplicates at the same well position on an ABI 7500 HT detection system (Applied Biosystems, Foster City, CA, USA). Each 20 µL reaction consisted of 20 ng DNA, 1 × SYBR Green master mix (Applied Biosystems) and 200 nM forward telomere primer (CGG TTT GTT TGG GTT TGG GTT TGG GTT TGG GTT), 200 nM reverse telomere primer (GGC TG TCT CCT TCT CCT TCT CCT TCT CCT TCT CCT). We used the following primer pairs for housekeeping gene analysis: 36B4u, CAG CAA GTG GGA AGG TGT AAT CC; 36B4d, CCC ATT CTA TCA TCA ACG GGT ACA A. The DNA quantum standard was verified using one reference sample diluted to final concentrations: 0.02, 0.20, and 2.00 ng/μL. The cycling of each qPCR analysis (for both telomere and housekeeping gene) started with a ten-minute cycle at 95 °C, followed by 45 15-s cycles at 95 °C and ended with a one-minute cycle at 60 °C. We generated both standard and dissociation curves using an Applied Biosystem Prism 7500 SDS software (Thermo Fisher Scientific). We performed the qPCR analysis with no independently cultivated cell lineage replicates (no biological replicates).

### 2.5. Immunocytochemistry

Using immunocytochemistry, we also detected the presence of the telomerase enzyme in analyzed DPSCs. For this analysis, DPSCs from the 2nd passage were seeded in a concentration of 5000 cells per cm^2^ and cultivated in chamber slides (Nalge Nunc International Corporation, Rochester, NY, USA) for two days. Before immunostaining, the adherent cells were fixed using 10% formaldehyde. After a thorough washing with phosphate-buffered saline (PBS), a 0.5% solution of Triton (250 mL Triton (Sigma-Aldrich) and 500 mL PBS) was used for 10 min to facilitate antibody penetration. Following a 20 min incubation of the sections with PBS containing a goat serum (1:20; Jackson ImmunoResearch Labs, West Grove, PA, USA), they were treated with a primary liquid mouse monoclonal antibody NCL-L-hTERT aimed against human telomerase reverse transcriptase (1:50; Novocastra, Leica Biosystems, Nußloch, Germany) for 60 min. After washing, the sections were treated with the secondary anti-mouse antibody IgG2a2b2c. For immunostaining, the samples were treated with ExtrAvidin−Cy3 (Sigma-Aldrich). We counterstained cell nuclei with 4′-6-diamidino-2-phenylindole (DAPI, Sigma-Aldrich) for 5 min and observed samples with a BX51 Olympus microscope. Images were overlapped using Adobe^®^ Photoshop CC 2020 (Adobe Systems, San Jose, CA, USA).

### 2.6. Differentiation Potential

We had already determined the ability of fresh, non-cryopreserved lineages to differentiate into chondrogenic and osteogenic cell lines [12]. We triggered osteogenesis and chondrogenesis in cells harvested from the 4th passage. We used the same differentiation protocols as in our previous study. Briefly, chondrogenesis was induced using the Differentiation Basal Medium-Chondrogenic (Lonza, Basal, Switzerland) enriched with 50 ng/mL TGF-β1 (R&D Systems, Minneapolis, MN, USA) for 21 days. Osteogenic differentiation was triggered using the Differentiation Basal Medium-Osteogenic (Lonza) also for three weeks. The medium was changed every three days.

Both osteogenic and chondrogenic differentiation was assessed via immunohistochemistry and histological staining. After induced chondrogenic differentiation, we visualized the type II collagen in the samples using a primary mouse IgM antibody (1:500, Sigma-Aldrich) and Cy3^TM^-conjugated goat anti-mouse secondary IgM antibody. The cell nuclei were counterstained with 4′-6-diamidino-2-phenylindole (DAPI, Sigma-Aldrich). In differentiated osteogenic samples, we quantified an osteocalcin using a primary mouse IgG antibody (1:50, Millipore, Burlington, MA, USA) and donkey anti-mouse secondary IgG antibody (1:250, Jackson ImmunoResearch Labs). We also stained paraffin osteogenic samples using the von Kossa stain to visualize calcium deposits as black spots in the produced extracellular mass.

We also tested the immunocytochemistry and histological staining on the opposite differentiated phenotypes than they are supposed to be used. Furthermore, we detected the osteocalcin and type II collagen in non-differentiated cell samples as negative controls.

### 2.7. Statistical Analysis

All statistical analyses were performed using the statistical software GraphPad Prism 6 (San Diego, CA, USA). The group data are presented as the medians. The statistical significance (*p* < 0.05) was determined using the Wilcoxon matched-pair analysis. The correlation between the change in relative telomere length and selected parameters (PDs and PDT) was tested using Pearson’s correlation analysis.

### 2.8. Following Study Supporting the Hypothesis of Time-Dependent Telomerase Activity

To support the hypothesis that the excessive in vitro cultivation might lead to a decrease of replicative capacity or, ultimately, stem cell exhaustion because of lack of time to compensate telomere loss by telomerase, we performed an additional subsequent study.

We included the same lineages as we had in the major project, but these were cryopreserved in the 7th passage using the uncontrolled rate freezing (Z05/p7, Z06/p7, Z08/p7). After cryopreservation, they were thawed using the same standard conditions, the 37 °C thermal bath. Upon thawing, DPSCs were split into two cultivation dishes, and we mirrored the cultivation protocol. Thawed DPSCs were expanded in the modified cultivation media for mesenchymal adult progenitor cells. One cultivation dish containing DPSCs was used for the phenotype analysis using the flow cytometry analyzer (Cell Lab Quanta, Beckman Coulter). The method is already described above. However, we analyzed the limited number of CD markers (CD29, CD31, CD34, CD44, CD45, C73, CD90) to confirm the mesenchymal origin of DPSCs. The second cultivation dish was split into four cultivation dishes and further used for relative telomere length measurement. One cultivation dish served as the control set where DPSCs were cultivated until they reached 100% confluence. Other telomere measurements were performed one, two, and three weeks after the cultivation with no passaging. We changed the medium every three days. The relative telomere length was measured using qPCR, the same protocol described above. Before each telomere measurement, we also determined the cell viability by trypan dye exclusion method using the Vi-Cell analyzer (Beckman Coulter).

We wanted to support or refute the hypothesis that telomerase compensatory activity might be efficient if it had enough time to compensate for telomere loss during ex vivo DPSC expansion.

## 3. Results

### 3.1. DPSC Lineages and Phenotype Analysis

We used ten lineages of DPSCs previously characterized, amplified, and cryopreserved in the 1st passage. These were stored at a temperature of −80 °C and thawed using a 37 °C thermal bath. We indicated them by a letter B in order to distinguish them from the non-cryopreserved lineages. Following the minimal classification criteria, stem cells expressed the CD markers for mesenchymal stem cells highly and showed low or no expression of CD markers for endothelial of hematopoietic stem cells. Specifically, they showed high expression (>70%) for surface mesenchymal stem cell markers CD29, CD44, CD73, and CD90. We determined no expression (<10%) of a hematopoietic precursor cell marker CD34. The hematopoietic cell surface marker CD45 and an adhesive molecule for endothelial cells, marker CD31, were lowly expressed (10–40%). We observed the moderate expression of CD105 (40–70%). Table 1 illustrates the median percentage of CD marker expressions of cryopreserved DPSCs in %.

### 3.2. DPSC Kinetics

Figure 1 depicts the DPSC kinetics of the DPSCs.

We determined DPSC kinetics according to their cumulative population doublings (PDs) and population doubling time (PDT) in hours. These observations gave us evidence of an increasing number of cumulative population doublings until the 8th passage and the extended time needed for the lineages to double. The median PDT was 58.8 h in the 8th passage (p8), and median PDs was 44.6 (from primary passage to 8th passage). After thawing in the thermal bath, cryopreserved DPSCs needed an extended time to reach established confluence in the 2nd passage.

### 3.3. Quantitative Polymerase Chain Reaction (qPCR) Analysis

In this study, we aimed to evaluate the effect of in vitro cultivation passaging on the relative telomere length in accordance with DPSC proliferation capacity. We evaluated the relative telomere length change between cells harvested from the 2nd passage and 7th passage as calculated using the formula T/S = 2^−ΔCt^. Table 2 illustrates T/S and standard deviations, PD, and PDT values from all ten lineages of DPSCs. Standard deviations of the T/S ratio were calculated for three technical replicates of each sample without biological replicates.

DPSCs lines showed an overall decrease in the relative telomere length with increasing passage number. However, the trend was found very narrowly below the statistical significance (*p* = 0.0502). In lineage Z01B, Z06B, Z07B, we observed the relative telomere length prolongation. In other words, the compensatory mechanism of telomerase was efficient, despite the increased proliferation demands on DPSCs throughout in vitro cultivation.

The higher the T/S values, the higher the number of reached PDs and shorter PDT were observed in the 7th passage. The correlations were found statistically significant (*p* for PDs = 0.018; *p* for PDT = 0.035). The linear regression line best fit showed data (Figure 2).

The relative telomere length attrition versus lower count of PDs and longer PDT among each DPSC lines were also found to be statistically significant (Figure 3).

Interestingly, the relative telomere length of some cells analyzed in the 2nd passage was longer than the figures of non-cryopreserved cells. The median of relative telomere length of non-cryopreserved cells was 3.6 [12], whereas for cryopreserved cells, it was 4.7. The correlation between the prolonged PDT and relative telomere length seen in p2 was statistically significant (*p* = 0.012), as seen in Figure 4.

### 3.4. Immunocytochemistry

We detected the presence of the human telomerase reverse transcriptase inside the DPSCs. After immunostaining, the telomerase was revealed as red areas inside the blue nuclei of DPSCs (Figure 5).

### 3.5. Differential Potential of DPSCs

We exposed DPSCs to conditions that triggered osteogenic and chondrogenic differentiation. After 3–4 weeks of differentiation, cells produced chondrogenic and osteogenic extracellular mass. Immunohistochemical staining proved type II collagen in chondrogenic differentiated and the osteocalcin in the differentiated osteogenic samples. The von Kossa histological staining revealed the calcium phosphate deposits as black spots in produced osteogenic extracellular mass. The figures illustrating the histological and immunocytochemical detection of DPSC multipotency are enclosed in supplementary data as Appendix A.

### 3.6. Subsequent Study Supporting the Hypothesis of Time-Dependent Telomerase Activity

In order to support or refute the hypothesis that the proliferation demands on isolated DPSCs cultivated in vitro exceed the telomerase capacity to compensate for the telomere loss, we did another subsequent study. We included the same lineages as we had in the major project, but these were cryopreserved in the 7th passage using the uncontrolled freezing (Z05/p7, Z06/p7, Z08/p7). We analyzed the reduced number of CD markers to confirm the mesenchymal origin. Table 3 illustrates the median percentage of CD marker expressions of cryopreserved DPSCs.

The same qPCR assay determined the relative telomere length measurement and calculation of the formula T/S = 2^−ΔCt^. One analysis was performed when DPSCs reached 100% confluence, and they served as the control group. Other measurements followed after one, two, and three weeks. Furthermore, we also determined cell viability before each analysis. The figure depicting the cell viability is enclosed in the supplement data as Appendix A. Figure 6 shows the T/S ratio and standard deviations for three technical replicates calculated in the control group and after one, two, and three weeks of cultivation with no passaging.

We observed the relative telomere length prolongation of two DPSC lineages (Z05/p7, Z08/p7) up to the analysis performed after two weeks. In contrast, we observed relative telomere length attrition in the final analysis correlated with cell viability drop. In the case of lineage Z06/7p, we did not observe telomere attrition or prolongation throughout the measurements.

## 4. Discussion

DPSCs belong among the mesenchymal adult stem cells, and they have a high potential to be used in regenerative and reparative therapies due to their remarkable features, such as self-renewal ability and multipotency. The easy accessibility of DPSCs from the dental pulp is a key advantage over other mesenchymal stem cells. However, a major drawback to using DPSCs is that they are only present in low numbers in the dental pulp. Therefore, the necessary amplification after their isolation and serial passages of the cells ultimately lead to a replicative senescence, limiting the number of available cells that could be used in the clinical setting. The extensive in vitro proliferation resembles the cell aging in vivo. The major populations of somatic cells do not express telomerase. Their replicative ability is restricted to the length of telomeric repeats at the end of their chromosomes [15]. If cells divide via mitosis at some point, the progeny reaches the Hayflick’s limit, which is believed to be 50 ± 10 cell divisions, and cells become senescent, and cell division stops.

Although the DPSCs have the advantage over somatic cells because they express the specific ribonucleoprotein enzyme, telomerase, which contracts the DNA loss, the high demands of proliferation during in vitro cultivation limit this compensatory mechanism. Mokry and al. [11] concluded that extensive in vitro proliferation of human DPSCs was associated with telomere attrition with significant correlations to prolongations of DPSCs doubling time. On the other hand, such remarkable expansion might be necessary for potential stem cell treatment in patients.

In this study, we wanted to evaluate the effect of cultivation passaging on the DPSC relative telomere length in accordance with proliferation capacity. We measured the relative telomere length using qPCR. Unlike other genes, the telomeres are not formed by a unique sequence of nucleotides in the whole length, but they are repeated sequences of six nucleotides. The number of these sequences is continuously reduced during life. The limitation of qPCR in this situation is that primers cannot recognize only the telomere’s start and ending sequence, but they bind randomly throughout the telomere sequence. Therefore, qPCR analysis in this situation is that longer telomeres increase the change for a generation of longer DNA sequences or their increased amount. The signal intensity is then compared to the highly conserved gene with stable length to standardize between measurements. This approach was repeatedly used and scaled in the literature [11,14,16].

All characteristics of unfrozen cells had already been published in our previous article [12]. Hence, we provided the basic characteristics of cryopreserved cells in this article. They highly expressed surface markers for mesenchymal stem cells (CD29, CD44, CD73, and CD 90). We determined no expression of hematopoietic marker CD34. The hematopoietic marker CD45 (protein tyrosine phosphatase, receptor type C) was lowly expressed. The higher expression is due to ITS’s presence in the cultivation medium, which keeps the cell more undifferentiated [17]. The same increased expression had already been seen in unfrozen cell lineages [12]. We did not confirm the data published by Alraies and col. They identified that the proliferative and regenerative heterogeneity was related to contrasting CD271 expression [18]. All DPSC populations of recent study lowly expressed CD271 in both 3rd and 7th passages.

Multipotency is another property of adult stem cells. We documented that cryopreserved DPSCs had chondrogenic and osteogenic potential under suitable differentiation media. We confirmed our results with immunohistochemistry and histological staining. We did not try to trigger adipogenesis because we were unsuccessful with the unfrozen cells, even though we had used the standard protocol and adipogenic differentiation medium. This result is not surprising because adipocytes are not a physiological component of dental pulp tissues in contrast with the bone marrow. Those observations correspond with Gronthos’s study findings, where they concluded that DPSCs differentiated in adipocytes unwillingly [19].

The DPSCs remained proliferatively active until p8, when we terminated all lineages. We demonstrated the decreasing cumulative population doublings and increasing population doubling time with increasing passaging. By the passage 7th, the cells reached the median of 42.1 population doublings, and the median of the population doubling time was 46.4 h. Although we visualized the presence of telomere reverse transcriptase in the nuclei of cryopreserved DPSCs using immunohistochemistry, we documented the relative telomere shortening with increasing cell passaging. The difference of relative telomere length between the 2nd and 7th passage, respective T/S ratio, was found slightly below the statistical significance. A limited number of cell lineages can explain it. We identified that proliferation rate change is closely related to the relative telomere length change. DPSCs with the higher T/S ratio reached a higher number of population doublings for shorter population doublings time. The higher attrition of relative telomere length (evaluated as the difference between T/S ratio for the 2nd passage and 7th passage), the lower number of cumulative PDs and longer PDT were determined in the 7th passage.

The lineages Z02B, Z03B, and Z04B slowed down their proliferation rate during cultivation. It was demonstrated that DPSCs are a heterogeneous group of cells varying in many biological features, and telomerase quantity or activity seems to be one of them. The various telomerase activity was already discussed in a previous study [15]. In these lineages where we observed a faster proliferation rate, we hypothesize that the quantity of telomerase or its activity rate was higher initially, and therefore they proliferated with no need of extra time to compensate for the telomere loss. On the other hand, the lineages with a slower proliferation rate exhausted the telomere repeats earlier during excessive in vitro cultivation. They needed more time to compensate for the telomere loss. More impressive was the prolongation of the relative telomere length of DPSCs seen in the 2nd passage compared to fresh cells [12]. DPSCs used in the study were thawed using a 37 °C thermal bath, and they required a longer population doubling time immediately after thawing. This time is necessary for cells to recover their “functional” health. During this time, they deal with the negative consequences of cryopreservation and thawing. During this time, DPSCs were cultivated with no passaging. Al-Sagi et al. observed a 2-day delay in recovering cryopreserved hematopoietic and adipose stem cells [20]. The documented prolonged relative telomere length opened the hypothesis that if the telomerase had had an adequate time to counteract the DNA loss, the compensatory mechanism would have been sufficient. In order to support or refute this theory, we did an additional subsequent study, including a reduced number of DPSC lineages. We determined the relative telomere length when DPSCs reached 100% confluence (served as the control group) and other analyses after one, two, and three weeks of cultivation with no passaging. We observed the relative telomere length prolongation in two lineages after 2-week cultivation with no passaging. In the third week of analysis, the relative telomere length attrition correlated with the cell viability reduction. We assumed that DPSCs had to proliferate to replace dead cells that were washed out during medium exchange. In the case of Z06/p7, we observed neither the telomere attrition nor prolongation. This lineage was one of such where we observed the prolongation of the relative telomere length between p2 and p7 in the main study. According to the main study and additional study, there seem to be two overlapping effects reflecting the proliferation rate of DPSCs. First, the telomere length and telomerase activity should be considered in the characterization of DPSCs, and second the necessary time for telomere prolongation during in vitro cultivation.

To our knowledge, this article is the first to report such results, but we are fully cognizant of the limitations of the main study as well as the additional subsequent study. Further examination should be performed before our observation could be generalized. In our future research, we would like to hypothesize further if the telomerase activity is time-dependent and quantify the telomerase in isolated DPSCs.

It was already described above that the preservation of stemness and self-renewal abilities of isolated stem cells remain challenging for researchers today. We speculate that one mechanism of preserving the relative telomere length in cells with high proliferation demands is to let cells cultivate longer to provide the telomerase enough time to transcribe the missing telomere repeats. According to the additional study, the sufficient time of cultivation with no passaging could be one to two weeks because we observed the unwilling reduction in the cell viability after three-week cultivation.

Another limitation of our studies is that we did not rule out the other possibilities to maintain the telomere length besides the telomerase activity. It has been reported that a recombination-based DNA replication mechanism may maintain telomere length [21]. In addition, sub-telomeric hypomethylation facilitates telomere elongation in mammalian cells suggesting an epigenetic modification of chromatin [22]. We visualized the telomerase in nuclei of DPSCs using antibody-based immunofluorescence detection. We are fully aware of the limitation of this detection.

We eliminated the chronological age-related heterogeneity among donors. All donors belonged to the age group of 13–18 years. We also excluded the possible cultivation effects on the relative telomere length by cultivating all DPSC lineages under the same cultivation conditions.

The telomerase activity levels differ among stem cells according to their origin. Embryonic stem cells highly express telomerase, making their replicative capability stable even during long-term cultivation in vitro conditions. In adult stem cells, the effect of telomerase is not entirely sufficient [23]. Hence, the telomere length and telomerase activity are some of the markers that should be considered during stem cell characterization, especially to characterize their replicative and proliferative capabilities.

## 5. Conclusions

We concluded that excessive proliferation demands on DPSCs during in vitro cultivation result in telomere attrition. However, we also observed the heterogeneity in compensatory mechanism activities of DPSCs, because three lineages did not experience the telomere shortening even in such extreme cultivation conditions. In these cell lines, we hypothesize that the telomerase activity or quantity was higher initially. Furthermore, we observed that the compensatory mechanism of telomerase activity might be time-dependent. By performing the additional subsequent study, we supported this hypothesis. It is necessary to amplify the number of isolated stem cells for potential usage in regenerative or reparative therapies. This necessary amplification after isolation and several passages leads to telomere attrition. In this study, we opened the theory that if we keep the stem cells in vitro cultivation with no passaging, the telomerase will have enough time to compensate for the telomere loss. Therefore, during the necessary in vitro amplification of DPSCs, it might be beneficial to keep the DPSC cultivating longer with no passaging to preserve the telomere length. However, in our future research, we would like to analyze this observation in more detail.

## Figures and Tables

**Figure 1 biomolecules-11-00464-f001:**
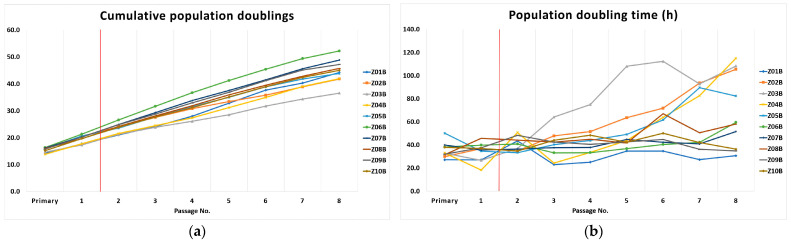
Kinetics of the Dental pulp stem cells (DPSCs). A red vertical line illustrates the time of DPSC cryopreservation in the p1: (**a**) Cumulative population doublings; (**b**) Population doubling time in hours.

**Figure 2 biomolecules-11-00464-f002:**
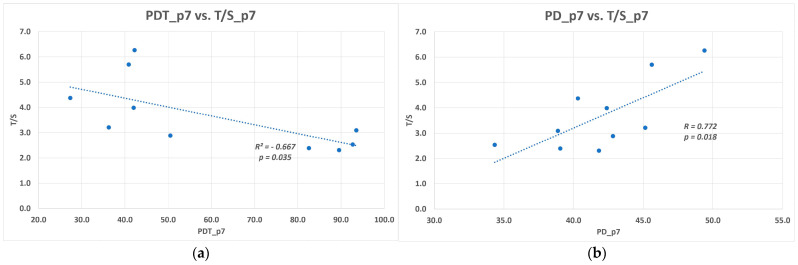
The relationship between the relative telomere length (T/S) and DPSCs kinetics in p7. The correlation coefficients of a linear regression fit the shown data the best: (**a**) Plot of the (T/S) in p7 against doubling time (PDT) in the p7, and the regression line; (**b**) Plot of T/S in p7 against cumulative population doublings in the p7, and the regression line.

**Figure 3 biomolecules-11-00464-f003:**
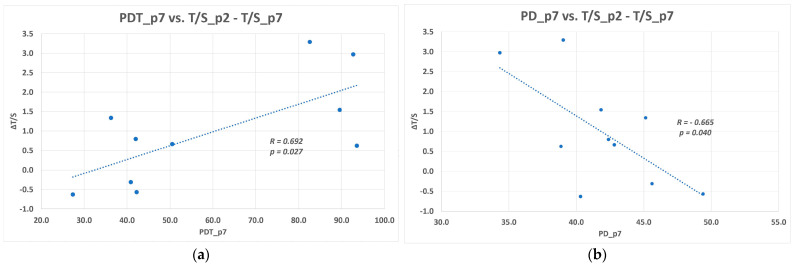
The relationship between the relative telomere length shortening (ΔT/S) between p2 and p7 and DPSC kinetics in p7. The correlation coefficients of a linear regression fit the shown data the best: (**a**) Plot of the ΔT/S against doubling time (PDT) in the p7, and the regression line; (**b**) Plot of the ΔT/S against cumulative population doublings (PDT) in the p7, and the regression line. The correlation coefficients of a linear regression fit the shown data the best.

**Figure 4 biomolecules-11-00464-f004:**
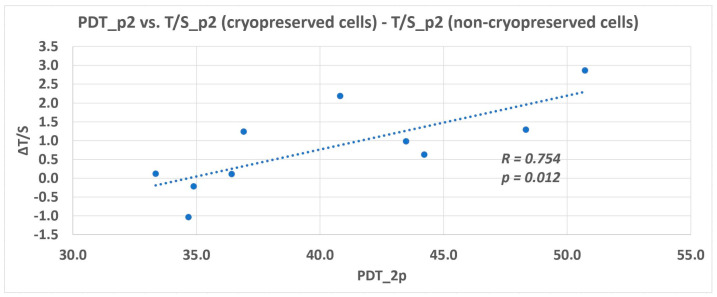
Cryopreserved cells had a significantly prolonged population doubling time (PDT) and relative telomere length in the p2 compared to non-cryopreserved cells. Plot of the difference between the relative telomere length of cryopreserved cells and non-cryopreserved cells in the p2 (ΔT/S) against the difference between the reached doubling time (DT) of cryopreserved cells and non-cryopreserved cells in the p2. The correlation coefficient of a linear regression fits the shown data the best. The regression line is illustrated.

**Figure 5 biomolecules-11-00464-f005:**
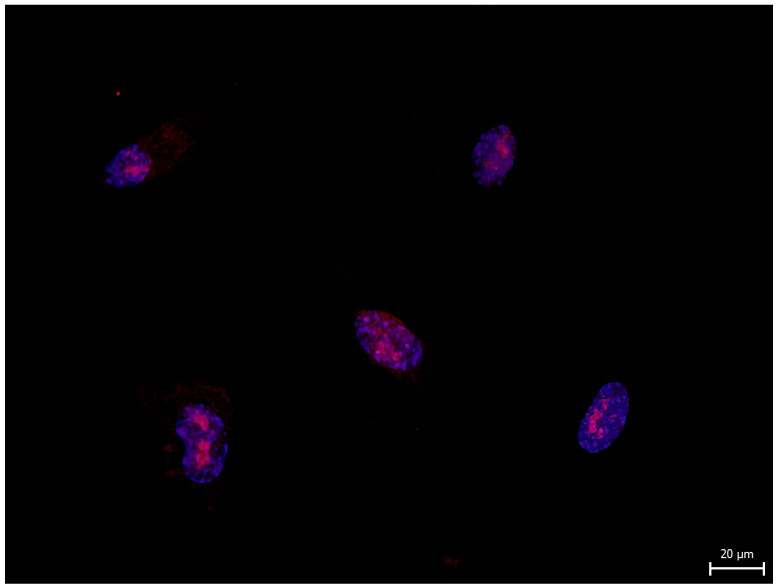
Human telomerase reverse transcriptase (hTRT, red areas) is expressed inside the nuclei (blue) of DPSCs from the 2nd passage. Scale bar 20 µm.

**Figure 6 biomolecules-11-00464-f006:**
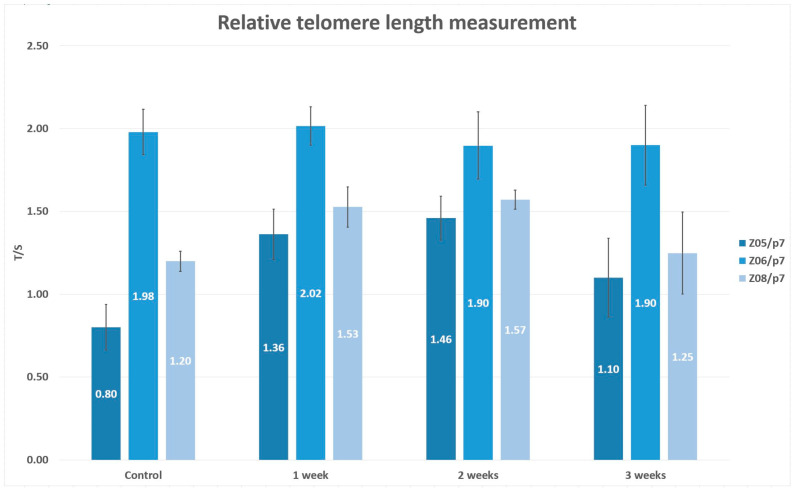
The relative telomere length measuremen—T/S ratio and standard deviations (lineages Z05/p7, Z06/p7, Z08/p7). The error bars were calculated for three technical replicates without biological replicates.

**Table 1 biomolecules-11-00464-t001:** The percentage of cells positively expressing tested cluster of differentiation (CD) markers. Data are shown as medians.

CD Markers	DPSCs (Z01B–Z10B)	
	p3	p7
CD29	96.8	96.7
CD31	30.2	16.4
CD34	0.7	0.5
CD44	87.4	94.9
CD45	21.9	11.9
CD73	94.8	95.0
CD90	94.5	97.7
CD105	58.3	58.2
CD271	8.6	11.3

Positive cells were determined as the percentage with a fluorescence intensity greater than 99.5% of the negative isotype immunoglobulin control.

**Table 2 biomolecules-11-00464-t002:** Telomere repeat copy number to single-gene copy number (T/S) ratio and standard deviations, cumulative population doubling (PD), and population doubling time (PDT) values from all ten lineages of DPSCs. Standard deviations of the T/S ratio were calculated for three technical replicates without biological replicates.

Lineage	T/S		PD		PDT	
	p2	p7	p2	p7	p2	p7
Z01B	3.74 ± 0.14	4.37 ± 0.3	21.0	40.3	43.5	27.3
Z02B	3.72 ± 0.19	3.09 ± 0.02	24.0	38.9	34.9	93.5
Z03B	5.50 ± 0.22	2.53 ± 0.14	21.2	34.3	36.9	92.7
Z04B	5.68 ± 0.12	2.39 ± 0.05	21.5	39.0	50.7	82.6
Z05B	3.85 ± 0.23	2.31 ± 0.21	23.6	41.8	33.3	89.6
Z06B	5.69 ± 0.28	6.26 ± 0.33	26.6	49.4	40.8	42.2
Z07B	5.39 ± 0.32	5.70 ± 0.04	24.9	45.6	36.4	40.9
Z08B	3.55 ± 0.01	2.88 ± 0.16	24.2	42.8	44.3	50.5
Z09B	4.55 ± 0.37	3.21 ± 0.22	24.8	45.1	48.3	36.2
Z10B	4.78 ± 0.06	3.99 ± 0.31	23.9	42.4	34.7	42.0

**Table 3 biomolecules-11-00464-t003:** Median percentage of cells positively expressing tested CD markers.

CD Markers	DPSCs (Z05/p7, Z06/p7, Z08/p7)
CD29	97.7
CD31	2.8
CD34	2.2
CD44	95.5
CD45	5.9
CD73	96.3
CD90	96.6

Positive cells were determined as the percentage with a fluorescence intensity greater than 99.5% of the negative isotype immunoglobulin control.

## Data Availability

Data is contained within the article or Appendix A.

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
