# Peer review of "The Effect of Cultivation Passaging on the Relative Telomere Length and Proliferation Capacity of Dental Pulp Stem Cells"

_biomolecules, 2021, doi:10.3390/biom11030464_

Round 1

Reviewer 1 Report

My concerns have been sufficiently addressed. Since the hypothesis is specifically based on telomere length with respect to passaging, a precisely defined passaging procedure is critical. The revised procedure included here helps. Greater details would help the readers even more.

Reviewer 2 Report

After the revision it became clear, that most of experiments in the article were done without biological replicates and can't be easily repeated by authors. Thus, overall quality of experimental data is low. 

But the key relationship between the relative telomere length shortening (ΔT/S) and population doublings looks interesting and enough reliable. 

Reviewer 3 Report

Nela Pilbauerova and colleagues performed studies, aiming to supply an experimental evidence supporting that cultivation passaging of dental pulp stem cells has negative effect on telomere length, and culturing these cells without passages leads to protection from telomere attrition.

Comments:

1. There is no enough experimental evidence from the presented studies to support the observation. It is not usually accepted to present in the research paper “adding pilot study” which is one of the main results of the study. This part should have comprehensive and complete investigation. It is important to show the schema of the experiment with all necessary details.

2. The manuscript is not well written and organized. English needs to be improved and edited by English native speaker. Paper should be majorly rewritten to present results clear and readable.

3. The materials and methods and result sections are not properly presented. Materials and methods should be short and comprehensive enough without explanations and discussion. In results should be explained comprehensively enough the aim, design and conclusion of the experiment. Results are poorly described.

4. “He (L. Hayflick) quantitatively examined the finite lifetime of diploid cells in vitro. Each cell within the population was endowed with the same doubling potential (50 ± 10).” This was made for differentiated somatic cells? Please explain and give reference how many cell divisions can make MSCs and DPSCs in cell culture conditions? To this regards it is important to extend the time of cultivation of DPSC lines beyond the 8 passage (44 cell divisions) in these experiments up to 12 passages. Population doublings is this number of cell divisions? Please explain points like these more clearly.

5. In methods, section 2.6: The authors indicated that the “DPSCs had not differentiated into adipogenic lines”. This mean that the DPSC lines are not full potential of multipotent cells. Are these cells true multipotent, it causes a big concern? There is no explanation or reference why these cells can not differentiate to adipogenic lineages?

6. The titles for each result section should be as a short conclusion. For example: not “3.4 Immunocytochemistry” but instead - Human telomerase reverse transcriptase (hTRT) is expressed in DPSCs. From what passage of these cells?

7. Figure legends is not comprehensive to understand the figures and results. The experiment and results are important to describe comprehensively to understand.

8. For Figure 1. needs to present control cells like differentiated cells – human fibroblasts (limited cell divisions), and human pluripotent cells (unlimited cell divisions).

9. In Figure 5. The negative and positive controls are absent. At what passage the cell staining was made?

10. In section 3.5. Differential potential of DPSCs – the results should be presented as main figure not in S2-4.

11. Supplementary figures should be properly composed and signed with figure legends.

Author Response

This manuscript is a resubmission of an earlier submission. The following is a list of the peer review reports and author responses from that submission.

Round 1

Reviewer 1 Report

The main task of the manuscript “The effect of cultivation passaging on the relative telomere length and proliferation capacity of dental pulp stem cells” is not devoted to telomeres or telomerase itself. Telomeres serves as a marker for optimization of conditions for stem cells amplification for large scale production via long-term cultivation. Thus the more specific for stem cells journal may have more relevant auditory for the manuscript (e.g. “Stem cells translational medicine” or “Stem cells and development” or other, devoted to stem cells problems).

There are few more obstacles for publication.

Major points:
- There is logically correct hypothesis, that “The documented prolonged telomere length opened the hypothesis that if the telomerase had had an adequate time to counteract the DNA loss, the compensatory mechanism would have been sufficient”. But cell lines Z01B, Z02B, Z06B with fast growth rate have longer telomeres then Z03B, Z04B, Z05B with slow growth rate after cultivation with passaging in main experiments. May be there are two overlapping effects: not only time for elongation, but also quantity of telomerase, which depends on cell line properties? Authors visualized the presence of telomere reverse transcriptase in nucleus, but it was qualitative experiment. Telomerase activity measurement at different passages looks much more valuable.
- Are less concentrations of growth factors in the media for lower growth rate may help better, then cultivation in monolayer, Are there no clone selection and population heterogenity after three weeks in monolayer?
- The manuscript contains several well-done (in accordance with experimental part of the manuscript) experiments, but some of them have no errors in shown results.
For example, there are no error bars on the figures 9 and 10? Moreover, QPCR assay, based on Cawton’s work is great for populations analysis, but is much less reliable for analysis of individual samples in comparison with Southern blot. Anyway, errors have to be shown.

Minor points
- Abbreviations “PD” and “PDT” are unclear in the abstract
- Experiments, devoted to differentiation of cells are aimed only to demonstrate the preservation of pluripotency and may be moved to supplement. Similarly, Fig 9 may be moved to supplement.
- Several mistyping, for example “aan adherent tissue” in row 97.
- Z02B and Z04B increases their growth rate during the work. Have authors an explanation of this effect?

Author Response

Reviewer_1

Dear reviewer,

We would like to thank you for reading our manuscript. We are grateful for all your comments and suggestions, which help us improve our research quality. Please, find below point by point authors' respond to all your comments.

The main task of the manuscript “The effect of cultivation passaging on the relative telomere length and proliferation capacity of dental pulp stem cells” is not devoted to telomeres or telomerase itself. Telomeres serves as a marker for optimization of conditions for stem cells amplification for large scale production via long-term cultivation. Thus the more specific for stem cells journal may have more relevant auditory for the manuscript (e.g. “Stem cells translational medicine” or “Stem cells and development” or other, devoted to stem cells problems).

Thank you for this comment. We believed that our manuscript is appropriate for this special issue, "Oral Regenerative Medicine: Current and Future." It is devoted to proliferation capacity and relative telomere length, and both topics are relevant to regenerative or reparative medicine. We agreed there are also other journals focusing on stem cell research with potential auditory. 

Major points

1) There is logically correct hypothesis, that “The documented prolonged telomere length opened the hypothesis that if the telomerase had had an adequate time to counteract the DNA loss, the compensatory mechanism would have been sufficient”. But cell lines Z01B, Z02B, Z06B with fast growth rate have longer telomeres then Z03B, Z04B, Z05B with slow growth rate after cultivation with passaging in main experiments. May be there are two overlapping effects: not only time for elongation, but also quantity of telomerase, which depends on cell line properties? Authors visualized the presence of telomere reverse transcriptase in nucleus, but it was qualitative experiment. Telomerase activity measurement at different passages looks much more valuable.

Thank you for this observation. We agreed with  the fact, that there are two overlapping effects. Lineages Z02B, Z03B, and Z04B slowed down their proliferation rate during cultivation. It was well demonstrated that DPSCs are a heterogeneous group of cells varying in many biological features, and telomerase quantity or activity seems to be one of them. The various telomerase activity was already discussed in the Jeon et al. study [15]. In these lineages where we observed a faster proliferation rate, we believe that the quantity of telomerase or its activity rate was higher initially, and therefore they proliferated with no need of extra time to compensate for the telomere loss.

On the other hand, the lineages with a slower proliferation rate exhausted the telomere repeats earlier during excessive in vitro cultivation. They needed more time to compensate for the telomere loss because the telomerase quantity or activity rate was lower. The discussion was extended with lines 391-399.

  1. Jeon, B.G.; Kang, E.J.; Kumar, B.M.; Maeng, G.H.; Ock, S.A.; Kwack, D.O.; Park, B.W.; Rho, G.J. Comparative analysis of telomere length, telomerase and reverse transcriptase activity in human dental stem cells. Cell Transplant. 2011, 20, 1693-1705, doi:10.3727/096368911X565001.

2) Are less concentrations of growth factors in the media for lower growth rate may help better, then cultivation in monolayer, Are there no clone selection and population heterogeneity after three weeks in monolayer?

The growth factors are added to the cultivation medium to lower the concentration of fetal bovine serum (FBS). The standard cultivation medium for progenitor adult stem cells contains 10 % of fetal bovine serum. We focused on decreasing FBS concentration in our previous study [1]. We concluded that DPSCs cultivated in the medium containing 2 % FBS, growth factors, and ITS (the same is used in the presented study) had the highest proliferation rate than those cultivated in medium with 10 % FCS and no growth factors. Plus, DPSCs cultivated in media containing 10% of FCS had significantly more chromosomal aberrations than those cultivated in media containing only 2% FCS and ITS.

It was already mentioned above. The DPSCs are heterogenous group of cells varying in many biological features. In our previous study [2], we isolated DPSCs and characterized them during in vitro cultivation up to the 8th passage. In order to answer your question about stem cell selection during in vitro cultivation in monolayer, we performed an additional statistical analysis. We observed distinctive differences in size, viability, or CD marker expression between the beginning and end of the cultivation, but these differences were not statistically significant (p<0.5).

  1. Suchanek, J.; Kleplova, T.S.; Kapitan, M.; Soukup, T. The effect of fetal calf serum on human dental pulp stem cells. Acta Medica (Hradec Kralove) 2013, 56, 142-149, doi:10.14712/18059694.2014.9.
  2. Pilbauerova, N.; Soukup, T.; Suchankova Kleplova, T.; Suchanek, J. Enzymatic Isolation, Amplification and Characterization of Dental Pulp Stem Cells. Folia Biol. (Praha) 2019, 65, 124-133.

3) The manuscript contains several well-done (in accordance with experimental part of the manuscript) experiments, but some of them have no errors in shown results.
For example, there are no error bars on the figures 9 and 10? Moreover, qPCR assay, based on Cawton’s work is great for populations analysis, but is much less reliable for analysis of individual samples in comparison with Southern blot. Anyway, errors have to be shown.

Figures 9 and 10 illustrate the values of three particular lineages. Data are not shown as mean or median. Therefore, there are no error bars or wrinkles for the percentile range. We wanted to illustrate each lineage separately to show the relative telomere length prolongation in accordance with the viability. We think that it is better for data presentation. However, according to this comment, we added a summary bar showing the mean and standard errors.

We agreed with the second part of the reviewer's comment. However, in the previous works from other laboratories, a comparison of both qPCR and Southern blot was performed, and both methods demonstrated similar results [1,2,3]. We performed a series of tests to detect reproducibility and concentration/length dependency when preparing previously published works on the same type of cells but also using cell lines. When we compared problems with the variability of blot efficacy and hybridization and detection sensitivity of probes in Southern blots with qPCR limitations, enabling only relative length measurement, we decided to use the more sensitive qPCR method. We also compare pair-wise – the same samples between passages in our previous studies [4].

  1. Lai, T.P.; Wright, W.E.; Shay, J.W. Comparison of telomere length measurement methods. Trans. R. Soc. Lond. B Biol. Sci. 2018, 373, doi:10.1098/rstb.2016.0451.
  2. Behrens, Y.L.; Thomay, K.; Hagedorn, M.; Ebersold, J.; Henrich, L.; Nustede, R.; Schlegelberger, B.; Göhring, G. Comparison of different methods for telomere length measurement in whole blood and blood cell subsets: Recommendations for telomere length measurement in hematological diseases. Genes Chromosomes Cancer 2017, 56, 700-708, doi:10.1002/gcc.22475.
  3. Lin, K.W.; Yan, J. The telomere length dynamic and methods of its assessment. Cell. Mol. Med. 2005, 9, 977-989, doi:10.1111/j.1582-4934.2005.tb00395.x.
  4. Pilbauerova, N.; Soukup, T.; Suchankova Kleplova, T.; Suchanek, J. Enzymatic Isolation, Amplification and Characterization of Dental Pulp Stem Cells. Folia Biol. (Praha) 2019, 65, 124-133

Minor points
1) Abbreviations “PD” and “PDT” are unclear in the abstract

We removed the abbreviation and replaced them with full phases (page1, lines 20,21)

2) Experiments devoted to differentiation of cells are aimed only to demonstrate the preservation of pluripotency and may be moved to supplement. Similarly, Fig 9 may be moved to supplement.

We moved the required figures to the supplement section.

3) Several mistyping, for example “aan adherent tissue” in row 97.

The spelling mistake was corrected.

4) Z02B and Z04B increases their growth rate during the work. Have authors an explanation of this effect?

Thank you for this note. The diversity between DPSCs was already described above. Although cell lines Z03B and Z04B had a high T/S ratio at the beginning of the cultivation, they slowed down their proliferation. We observed a slower proliferation rate in these lineages, we believed that they exhausted the telomere earlier during excessive in vitro cultivation, and they needed the extra time to activate telomerase to compensate for the telomere loss. It also reflects the observed correlation between the relative telomere length and proliferation rate.

Best regards,

Authors

Reviewer 2 Report

The research described in this manuscript has the potential to reveal some interesting insights into the potential of dental pulp stem cells. The authors conclude that telomerase does not have sufficient time to extend the telomeres during rapid expansion. Therefore a period without passaging will expand the doubling capacity of these cells. That is a possible outcome, but it is not fully supported by the presented data. In addition there are some clarity issues with the presentation. 

The introduction has a good description of telomeres and senescence, but it does not establish why the recommendation that BMSC be used within 4-7 passages applies to DPSC.

The passage procedures lack some detail that is central to the way the data was analyzed and presented. Was the medium changed other at time of passage? How was 70% confluence quantified? Does it mean that every dish was passaged at separate times? Could small differences in when each was passaged have large effects on doubling time, since this is the hypothesis being tested?

If 70% confluence was strictly used for determining time of passage, why don't the cumulative doublings vs. passage number of all follow a single curve? Wouldn't they all have the same number of doublings exactly? Any variation would be due to error in selecting the time of reaching 70% confluence.

The critical parameter T/S ratio is poorly defined. What does relative telomerase length mean? How does correlating it to PD or PDT infer proliferation capacity?

What is the hypothesis? Is it that these cells proliferate in vitro too fast for telomerase to keep up? Compared to in vivo? This comparison was not made. 

The jump to non cryopreserved cells in Figure 4 is confusing. What does the difference in T/S mean? Even though this will be discussed in a later section, explaining why is necessary here. 

Figure 5/L267 what is the importance of this data? It's best not to say "successfully" detect, it is either there or it is not. 

Figures 6-8: Was there a negative result for differentiation markers for the opposite phenotype? For example, was osteogenesis negative in chondrogenic differentiation?

What purpose or effect did the 12 month cryopreservation have?

In the Discussion and conclusion, the authors speculate that telomere length can be preserved by a no passaging period of 1-2 weeks. However, this is not tested. It cannot be predicted if proliferation rate or capacity will be restored after prolonged confluence. The conclusion on this seems overstated to say the hypothesis is confirmed.  

Author Response

Reviewer_2

Dear reviewer,

We would like to thank you for reading our manuscript. We are grateful for all your comments and suggestions, which help us improve our research quality. Please, find below point by point authors' respond to all your comments.

The research described in this manuscript has the potential to reveal some interesting insights into the potential of dental pulp stem cells. The authors conclude that telomerase does not have sufficient time to extend the telomeres during rapid expansion. Therefore, a period without passaging will expand the doubling capacity of these cells. That is a possible outcome, but it is not fully supported by the presented data. In addition, there are some clarity issues with the presentation.

Thank you very much for mentioning these comments. It indeed might be an exciting outcome. On the other hand, we are fully cognizant of the main study's limitations and additional pilot study. Further examination should be performed before our observation could be generalized. In our future research, we would like to hypothesize the theory in detail.

1) The introduction has a good description of telomeres and senescence, but it does not establish why the recommendation that BMSC be used within 4-7 passages applies to DPSC.

Indeed, we did not establish why this recommendation for BM-SCs would be valid for DPSCs. We just wanted to highlight that necessary stem cell amplification before potential application in stem cell therapies can lead to proliferation decline. We change the reference to one which is more accurate for hMSCs. We also change the introduction according to it. (page 2, line 61 - 65)

2) The passage procedures lack some detail that is central to the way the data was analyzed and presented. Was the medium changed other at the time of passage? How was 70% confluence quantified? Does it mean that every dish was passaged at separate times? Could small differences in when each was passaged have large effects on doubling time, since this is the hypothesis being tested?

The medium was changed every three days. We added this missing information in the Material and Method section (page, line).

Our research team has been studied dental-related stem cells for decades, and during that time, we have gained experience in the field. We estimated the 70 % confluency by comparing several representative pictures during the observation in the phase microscope. The confluence before each cell passage was always discussed among experts of our research team. The method was the same as it was in our previous studies [1-10].

  1. Mokry, J.; Soukup, T.; Micuda, S.; Karbanova, J.; Visek, B.; Brcakova, E.; Suchanek, J.; Bouchal, J.; Vokurkova, D.; Ivancakova, R. Telomere attrition occurs during ex vivo expansion of human dental pulp stem cells. J. Biomed. Biotechnol. 2010, 2010, 673513, doi:10.1155/2010/673513.
  2. Pilbauerova, N.; Soukup, T.; Suchankova Kleplova, T.; Suchanek, J. Enzymatic Isolation, Amplification and Characterization of Dental Pulp Stem Cells. Folia Biol. (Praha) 2019, 65, 124-133.
  3. Suchanek, J.; Kleplova, T.S.; Kapitan, M.; Soukup, T. The effect of fetal calf serum on human dental pulp stem cells. Acta Medica (Hradec Kralove) 2013, 56, 142-149, doi:10.14712/18059694.2014.9.
  4. Suchanek, J.; Visek, B.; Soukup, T.; El-Din Mohamed, S.K.; Ivancakova, R.; Mokry, J.; Aboul-Ezz, E.H.; Omran, A. Stem cells from human exfoliated deciduous teeth--isolation, long term cultivation and phenotypical analysis. Acta Medica (Hradec Kralove) 2010, 53, 93-99, doi:10.14712/18059694.2016.66.
  5. Suchánek, J.; Browne, K.Z.; Nasry, S.A.; Kleplová, T.S.; Pilbauerová, N.; Schmidt, J.; Soukup, T. Characteristics of Human Natal Stem Cells Cultured in Allogeneic Medium. Braz. Dent. J. 2018, 29, 427-434, doi:10.1590/0103-6440201802388.
  6. Pisal, R.V.; Suchanek, J.; Siller, R.; Soukup, T.; Hrebikova, H.; Bezrouk, A.; Kunke, D.; Micuda, S.; Filip, S.; Sullivan, G., et al. Directed reprogramming of comprehensively characterized dental pulp stem cells extracted from natal tooth. Sci. Rep. 2018, 8, 6168, doi:10.1038/s41598-018-24421-z.
  7. Suchanek, J.; Nasry, S.A.; Soukup, T. The Differentiation Potential of Human Natal Dental Pulp Stem Cells into Insulin-Producing Cells. Folia Biol. (Praha) 2017, 63, 132-138.
  8. Tuček, L.; Kočí, Z.; Kárová, K.; Doležalová, H.; Suchánek, J. The Osteogenic Potential of Human Nondifferentiated and Pre-differentiated Mesenchymal Stem Cells Combined with an Osteoconductive Scaffold - Early Stage Healing. Acta Medica (Hradec Kralove) 2017, 60, 12-18, doi:10.14712/18059694.2017.43.
  9. Suchánek, J.; Suchánková Kleplová, T.; Řeháček, V.; Browne, K.Z.; Soukup, T. Proliferative Capacity and Phenotypical Alteration of Multipotent Ecto-Mesenchymal Stem Cells from Human Exfoliated Deciduous Teeth Cultured in Xenogeneic and Allogeneic Media. Folia Biol. (Praha) 2016, 62, 1-14.
  10. Suchánek, J.; Soukup, T.; Ivancaková, R.; Karbanová, J.; Hubková, V.; Pytlík, R.; Kucerová, L. Human dental pulp stem cells--isolation and long term cultivation. Acta Medica (Hradec Kralove) 2007, 50, 195-201.

3) If 70% confluence was strictly used for determining time of passage, why don't the cumulative doublings vs. passage number of all follow a single curve? Wouldn't they all have the same number of doublings exactly? Any variation would be due to error in selecting the time of reaching 70% confluence.

It has been demonstrated since their first description by Gronthos et al. in 2000 that DPSCs are heterogeneous groups of cells varying in many biological properties (colony-forming units, cell size, plating efficiency, doubling time, surface markers). Therefore, it is generally recommended to perform biological replicates. The diversity between them can lead to distinctive differences. Therefore, it is unlikely that all isolated DPSCs in our manuscript would have acted the same even though we kept them under the same cultivation conditions.

4) The critical parameter T/S ratio is poorly defined. What does relative telomerase length mean? How does correlating it to PD or PDT infer proliferation capacity?

The qPCR assay performed telomere length measurement according to the method described by Cawthon with small modifications [1]. The T/S ratio represents the telomere repeat copy number (T) to the single-gene copy number (S). The T/S for each sample was normalized to the T/S value of a reference DNA sample to standardize between different runs. That is, – ΔΔCt was calculated for each sample. This value is proportional to the average telomere length of the evaluated sample. 36B4, encoding acidic ribosomal phosphoprotein, was used as the single-copy gene. We reflected it in the Material and Method section.

Thank you for these comments. We agree that closer reflection is necessary. Unlike other genes, the telomeres are not formed by a unique sequence of nucleotides in the whole length, but they are repeated sequences of six nucleotides. The number of these sequences is continuously reduced during life. The limitation of qPCR in this situation is that primers can not recognize only the telomere's start and ending sequence, but they bind randomly throughout the telomere sequence. Therefore, qPCR analysis in this situation is that longer telomeres increase the change for a generation of longer DNA sequences, or their increased amount. The signal's intensity is then compared to the highly conserved gene with stable length to standardize between measurements. This approach was repeatedly used and scaled in the literature [1,2,3].  The discussion was extended with lines 352 - 360.

The correlation between telomere length attrition and decreasing stem cell proliferation rate (PD, PDT) seen in our manuscripts has been described in previous studies [2,4,5].

  1. Cawthon, R.M. Telomere measurement by quantitative PCR. Nucleic Acids Res. 2002, 30, e47, doi:10.1093/nar/30.10.e47.
  2. Mokry, J.; Soukup, T.; Micuda, S.; Karbanova, J.; Visek, B.; Brcakova, E.; Suchanek, J.; Bouchal, J.; Vokurkova, D.; Ivancakova, R. Telomere attrition occurs during ex vivo expansion of human dental pulp stem cells. Biomed. Biotechnol. 2010, 2010, 673513, doi:10.1155/2010/673513.
  3. Lai, T.P.; Wright, W.E.; Shay, J.W. Comparison of telomere length measurement methods. Trans. R. Soc. Lond. B Biol. Sci. 2018, 373, doi:10.1098/rstb.2016.0451.
  4. Flores, I.; Benetti, R.; Blasco, M.A. Telomerase regulation and stem cell behaviour. Opin. Cell Biol. 2006, 18, 254-260, doi:10.1016/j.ceb.2006.03.003.
  5. Martínez, P.; Blasco, M.A. Telomere-driven diseases and telomere-targeting therapies. Cell Biol. 2017, 216, 875-887, doi:10.1083/jcb.201610111.
  6.  

5) What is the hypothesis? Is it that these cells proliferate in vitro too fast for telomerase to keep up? Compared to in vivo? This comparison was not made.

The comparisons of excessive proliferation in vitro and in vivo are impossible because adult mesenchymal stem cells in vivo are activated only during reparative or regenerative processes. Mostly, they are in a cell pool waiting for impulses to proliferate. On the other hand, to achieve positive results in potential clinical application, it is essential to get many therapeutic cells during long-term cultivation in vitro. DPSCs are forced to proliferated rapidly, and there is no time for telomerase to compensate for telomere loss and this excessive stem cell amplification can shorten the telomere to the critical point and reduce or block stem cell proliferation. Therefore, we hypothesized that if the telomerase had enough time to compensate for telomere loss caused by excessive cell amplification in vitro, would the telomerase compensatory effect be more effective. We did not want to compare in vitro and in vivo stem cell proliferation.

6) The jump to non cryopreserved cells in Figure 4 is confusing. What does the difference in T/S mean? Even though this will be discussed in a later section, explaining why is necessary here.

We agreed that figure 4 might be confusing, and it needs to be explained better. We added the explanation of what the difference of T/S means in the figure description.

7) Figure 5/L267 what is the importance of this data? It is best not to say "successfully" detect, it is either there or it is not.

We are confused about what L267 means but about the importance of immunocytochemical telomerase detection: according to Alraies et al. study, the telomerase presence in DPSCs can be doubted. Therefore, we qualify if the telomerase is presented in our isolated DPSCs.

We erased "successfully" in the text. (page, line)

7) Figures 6-8: Was there a negative result for differentiation markers for the opposite phenotype? For example, was osteogenesis negative in chondrogenic differentiation?

Thank you very much for this suggestion about how we can improve our future research. We did not try histological staining or immunocytochemical detection on different phenotypes than they are supposed to, but we performed the negative control on undifferentiated DPSCs. 

8) What purpose or effect did the 12 month cryopreservation have?

This information is not necessary for the outcome. It is a part of the description of used DPCSs, but we erased it according to the comment.

9) In the Discussion and conclusion, the authors speculate that telomere length can be preserved by a no passaging period of 1-2 weeks. However, this is not tested. It cannot be predicted if the proliferation rate or capacity will be restored after prolonged confluence. The conclusion on this seems overstated to say the hypothesis is confirmed. 

Thank you for these comments. Thanks to that, we are fully aware of this study's limitation; we did not want to conclude that cultivation with no passaging can delay the proliferative decrease. We pointed up one of the possible options for preserving the telomere length if the necessary stem cell amplification in vitro is needed. It might be beneficial to keep cells in more prolonged cultivation without passaging than passage them whenever they reached 70 % confluence. Therefore, we performed additional minor research to open this hypothesis. However, to conclude such as results, future research is necessary, and we hope we will be able to study this observation in detail this phenomenon soon and share our findings in one of the upcoming manuscripts.

Best regards,

Authors

Round 2

Reviewer 1 Report

I agree that the theme of this article is appropriate for a specialized issue ("Oral Regenerative Medicine: Current and Future.")

There are few obstacles for publication, because authors did not answer enough to the major points 1 and 2:

1a) Table 2, figure 6 and other data still do not contain error values.

Authors answered in the previous report, that “we added a summary bar showing the mean and standard errors.”. Adding of the average T/S ratio at the figure 6 does not answered to the questions about errors. The question about error bars was about each value on the histogram.

1b) How many biological replicates (independently cultivated replicates of each cell lines) were done for all experiments in the article? Have the experimental data been validated in independent replicates?

Authors wrote in telomere assay description, that “we analyzed each sample in triplicates”. If authors do not perform biological replicates (experiments with independently passaged cell lines) this should be described. This is due to significantly greater inter-experimental error for PCR-based assays than intra-experimental error. Corresponding notes “errors were calculated for three technical replicates without biological replicates” or “intra-experimental error is given” should be added to the data with errors.

For experiments without biological replicates it has to be mentioned in the text that the experiment was carried out without independent replication.

2) Adding data on telomerase activity in original cell lines (even without time-dependent experiments) could significantly improve the article. Otherwise, some statements should be corrected:

Line 390  “In these lineages where we observed a faster proliferation rate, we believe that the quantity of telomerase or its activity rate was higher initially, and therefore they proliferated with no need of extra time to compensate for the telomere loss. On  the other hand, the lineages with a slower proliferation rate exhausted the telomere repeats earlier during excessive in vitro cultivation. They needed more time to compensate for the telomere loss because the telomerase quantity or activity rate was lower.” Line 448 “In these cell lines, we believe that the telomerase activity or quantity was higher initially.”

The hypothesis is quite logical and authors may speculate about their assumptions. But the expression “I believe” is not the best in this context. Without measuring the telomerase activity, there are only assumptions, and this should be noted in the text as “speculation” or “hypothesis”

Minor point:

Line 433: authors wrote, that  “Furthermore, it is even unclear if telomerase is expressed in DPSCs [16]”, but earlier authors wrote “The various telomerase activity was already discussed in a previous study [15].”

Reviewer 2 Report

The response to my question about why PD is not similar for all samples is not convincing, and the authors confuse replicates with unique samples. There do not appear to be any replicates of each sample. If there were replicates done for each cell line, then error bars representing the replicates should be shown. The heterogeneity described would lead to greater within-sample variability, but not between-sample variability. If they were each passaged at 70% confluence, then they would each double the same number of times for the same number of passages because the passage time is adjusted to normalize doublings. There is no doubt biological variation between cell lines would cause differences in PDT, but only experimental error would lead to a systematic difference in number of PD. Biological variation does not explain why PD would be consistently higher for one line vs another if the cells were always passaged at 70%. When PDT is between 20 and 40 hr (for most data points here), the difference of just a few hours will have a huge effect on PD. Therefore, for this paper that uses this as the primary figure of merit, the authors should very clearly provide details on how the time of passage was specified (i.e., how was 70 % measured?) and how systematic errors could have been introduced. 

Fig 4: does 2p mean p2?

While opposite differentiation markers were not investigated, the authors responded that they included negative controls. Why isn't the phenotype negative control data for undifferentiated cells shown for differentiation studies if it was done?